# Colorimetric determination of urea using diacetyl monoxime with strong acids

**Noah James Langenfeld** ◎*, **Lauren Elizabeth Payne**◎, **Bruce Bugbee**◎

Department of Plants, Soils, and Climate, Crop Physiology Laboratory, Utah State University, Logan, Utah, United States of America

◎ These authors contributed equally to this work.
* noah.langenfeld@usu.edu

## Abstract

Urea is a byproduct of the urea cycle in metabolism and is excreted through urine and sweat. Ammonia, which is toxic at low levels, is converted to the safe storage form of urea, which represents the largest efflux of nitrogen from many organisms. Urea is an important nitrogen source in agriculture, is added to many industrial products, and is a large component in wastewater. The enzyme urease hydrolyzes urea to ammonia and bicarbonate. This reaction is microbially mediated in soils, hydroponic solutions, and wastewater recycling and is catalyzed *in vivo* in plants using native urease, making measurement of urea environmentally important. Both direct and indirect methods to measure urea exist. This protocol uses diacetyl monoxime to directly determine the concentration of urea in solution. The protocol provides repeatable results and stable reagents with good color stability and simple measurement techniques for use in any lab with a spectrophotometer. The reaction between diacetyl monoxime and urea in the presence of sulfuric acid, phosphoric acid, thiosemicarbazide, and ferric chloride produces a chromophore with a peak absorbance at 520 nm and a linear relationship between concentration and absorbance from 0.4 to 5.0 mM urea in this protocol. The lack of detectable interferences makes this protocol suitable for the determination of millimolar levels of urea in wastewater streams and hydroponic solutions.

## Introduction

Urea is a small organic compound used as the primary nitrogenous waste product in mammals. The protein content of a diet dictates the urea concentration, which is routinely measured in human blood serum as an indicator of healthy metabolism [1]. Urea is produced from the oxidation of amino acids and ammonia and is transported to the kidneys where it is used as a safe storage form of excess nitrogen. Urea is then excreted from the body in urine where it represents the largest concentration of any component aside from water [2]. Monitoring urea in wastewater can help guide bioreactor design and operation of downstream processing.

Urea is extensively used in agriculture as an inexpensive nitrogen fertilizer with more than 50% of all nitrogen applied as urea [3]. Urea hydrolysis to ammonia is catalyzed by urease, which is ubiquitous in many plants and microbes, but can cause alkaline substrate conditions and toxic levels of ammonia if not controlled [4]. Urea must be converted to ammonium either by soil microbes or plant derived urease before it can be assimilated into plant proteins.

**Data Availability Statement:** All relevant data is included in supplemental information files.

**Funding:** This research was supported by the Utah Agricultural Experiment Station, Utah State University, and approved as journal paper number

9459; NASA, Center for the Utilization of Biological
Engineering in Space (grant number
NNX17AJ31G). The funders had and will not have a
role in study design, data collection and analysis,
decision to publish, or preparation of the
manuscript.

**Competing interests:** The authors have declared
that no competing interests exist.

Monitoring urea concentration in soil leachate or liquid hydroponics helps quantify the hydrolysis rate which can lead to a basis for nitrogen application rates [5]. Quantification of urea is especially important in regenerative life support systems for long-term space missions to ensure efficient nitrogen recycling and recovery from urine [6].

Urea concentration can be determined indirectly via the products of hydrolysis or directly through several colorimetric methods [7, 8]. Ammonia and carbon dioxide are produced during the hydrolysis of urea by urease and can be measured gasometrically [9] or colorimetrically [10–12], respectively, to stoichiometrically determine the urea concentration. Indirect enzymatic measurements are sensitive to solution pH, divalent cation interference, and incomplete hydrolysis, which do not affect direct colorimetric determination [13]. Direct methods for urea determination complex urea with an aldehyde or ketone under strong acidic conditions to form a red to yellow colored product, which is then measured either colorimetrically or with liquid chromatography. Variations on the condensation reagent include the use of xanthydrol [14, 15], diacetyl monoxime [8, 16], dimethylglyoxime [17], or p-dimethylamino-benzaldehyde [18] which affect the color stability, reaction time, and sensitivity. Diacetyl monoxime is one of the more stable and easy to obtain reagents and is the focus of this protocol due to its fast reaction time with urea and chromophore intensity and stability [19] when reacted in the presence of acid, ferric chloride, and thiosemicarbazide.

Diacetyl monoxime breaks down into diacetyl during the reaction in the presence of heat (provided from a boiling water bath). Diacetyl and urea then condense in the same medium under the presence of a strong acid to form the yellow-colored diazine product and water. Diazine is light sensitive when sulfuric acid is used, but the addition of phosphoric acid [20] helps eliminate this sensitivity. Diazine is stabilized by thiosemicarbazide and converted to a pink-colored complex with a stronger absorbance in the presence of ferric ions derived from ferric chloride hexahydrate [21]. The mixed acid reagent is stable for at least a month at room temperature [22, 23], while the mixed color reagent is stable for at least a week. Reay [24] noted a decrease in response from the color reagent when allantoin and hydantoin were analyzed at micromolar levels, but no significant change was observed for urea concentrations. The maximum absorption of the final product is at 520 nm and is proportional to the concentration of urea (Fig 1) [25].

Diacetyl monoxime was first used by Fearon [26] as a test for citrulline, an alpha-amino acid and important intermediate of the urea cycle. Citrulline is a monosubstituted urea derivative and will give a positive result [27] if diacetyl monoxime is used to detect urea. This is not a factor if urea is being detected in wastewater streams as the concentration of citrulline [19] is nearly four orders of magnitude less than that of urea [28]. A similar concentration disparity exists in hydroponic solutions. Reay [24] found the diacetyl monoxime method overestimated urea concentrations in soil solutions by responding positively with many uredio compounds. This occurred when analyzed urea was at micromolar levels as opposed to the millimolar levels described in this protocol. The expected concentration range of samples must be assessed before determining suitability for this assay.

The average lower limit of detection (LoD) using this protocol is 440 μM urea (S2 File) using the calibration plot method described by Anderson [29]. Environmental samples typically have urea concentrations several orders of magnitude greater than this LoD. Reagent concentrations can be reduced if a lower LoD is desired [25], but are not the subject of this protocol. The method remains linear up to 5 mM urea, after which the absorbance exceeds 1 and the relationship is no longer linear. The average molar attenuation coefficient for this assay was 199 $\mu M^{-1}\, cm^{-1}$.

This protocol was developed specifically for the analysis of urea in wastewater and hydroponic solutions. The separation of urea from urine and its use as a fertilizer for plants is of

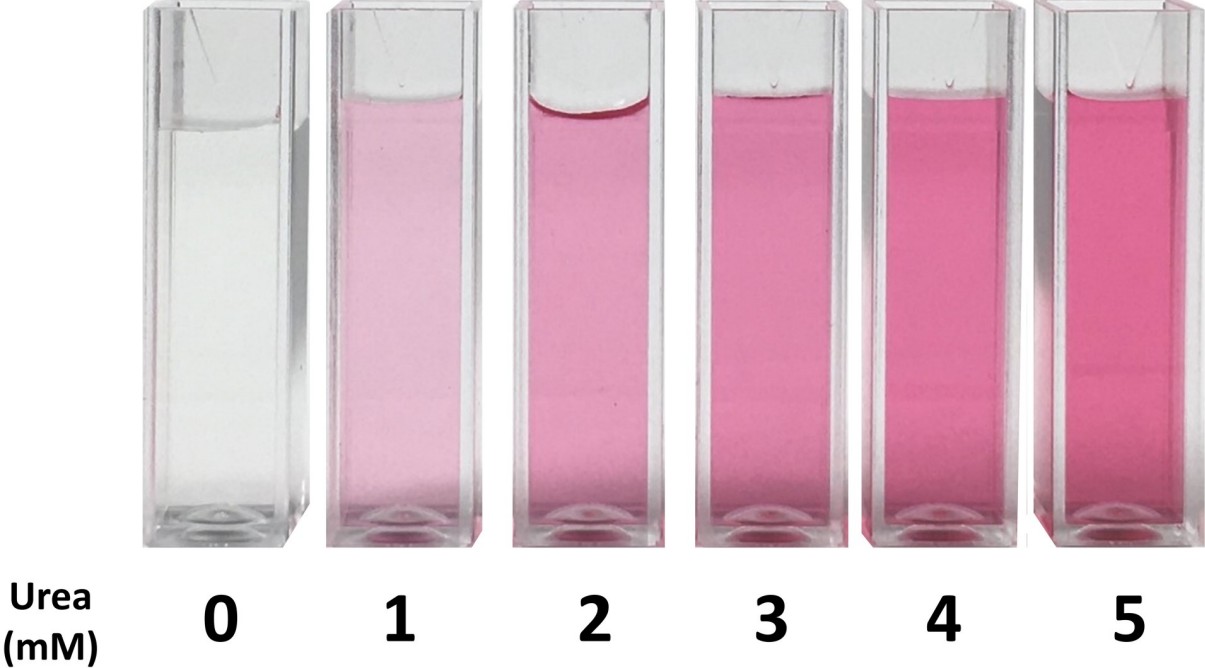

**Fig 1. Urea samples after absorbance measurement.** Diacetyl monoxime and urea produce a pink-colored complex in the presence of ferric ions with a maximum absorbance at 520 nm proportional to urea concentration.

special interest to the National Aeronautics and Space Administration for regenerative life support systems. Reagent concentrations, reaction time, and detection range in this protocol have been set to meet urea concentrations found in these scenarios. The protocol is repeatable and safe to perform if standard analytical procedures are followed. A step-by-step guide for both the reagent preparation and assay procedure are included to simplify the measurement process, minimize potential error, and obtain accurate measurements of urea concentrations at millimolar levels.

## Materials and methods

The protocol described in this peer-reviewed article is published on protocols.io, http://dx.doi.org/10.17504/protocols.io.byvipw4e and is included for printing as S1 File with this article.

## Expected results

Standards from 0 to 5 mM urea were prepared to generate calibration curves at 520 nm using a Shimadzu UV-2401PC (Shimadzu Corporation, Kyoto, Japan) spectrophotometer with a resolution of 0.1 nm and a path length of 1 cm. Calibration curves were constructed over 7 days using standards in triplicate. Mixed color reagent (diacetyl monoxime and thiosemicarbazide, MCR) stability was analyzed by constructing a set of calibration curves prepared with a MCR stored at 25 ˚C in the light and another set with a MCR stored at 4 ˚C in the dark. The relationship remained linear for both curve sets up to 5 mM urea over the course of the trial (Fig 2), after which the absorbance exceeded 1.000. The molar attenuation coefficient (202 $\mu M^{-1}$ $cm^{-1}$ at 25 ˚C and 196 $\mu M^{-1}$ $cm^{-1}$ at 4 ˚C) was calculated using the Beer-Lambert Law (A = $\varepsilon$bC), where A is the absorbance, $\varepsilon$ is the molar attenuation coefficient, b is the path length, and C is the concentration in mM Urea. The Beer-Lambert Law can then be used to determine the

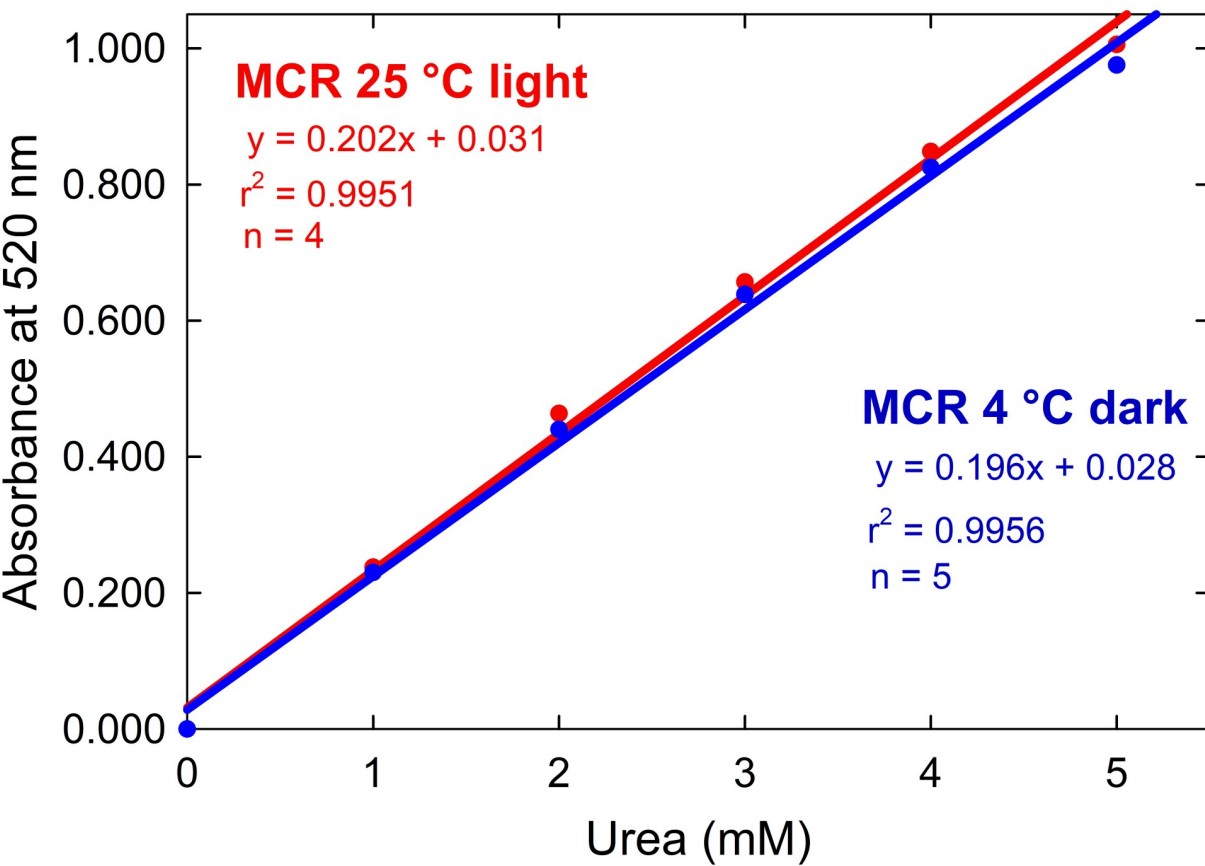

**Fig 2. Calibration curve for urea.** Calibration curves at 520 nm using a mixed color reagent (diacetyl monoxime and thiosemicarbazide, MCR) stored at 25 °C in the light and 4 °C in the dark. Error bars are too small to be shown (data included in S2 File). n = 4 for the 25 °C MCR and n = 5 for the 4 °C MCR.

concentration of unknown urea samples once ε is known. The average limit of detection was 0.455 mM urea for the 25 °C MCR and 0.425 mM urea for the 4 °C MCR (S2 File).

This protocol was applied in a practical setting to measure urea conversion to ammonium in a recirculating column system (Fig 3) over 30 days. A 2 mM urea (4 mM nitrogen) solution was made and transferred into recirculating columns. Each column was filled with perlite to provide a surface area for microbes and had automatic pH control to maintain pH below 7 and reduce volatile ammonia losses. The nitrogen concentrations contributed from ammonium and urea in each column were measured over 30 days using the Nesslerization colorimetric method [30] and this protocol, respectively. Results in Fig 4 show a decrease in urea and increase in ammonium as urea is hydrolyzed in the column. Total N decreased over the course of the study due to some volatilization of ammonia gas.

The protocol was also tested in a background of a hydroponic nutrient solution at pH 5.8 to simulate analyzing urea concentration when urea is used as a hydroponic nitrogen source. The solution prepared from reagent grade chemicals contained the following: 4 mM urea N, 2 mM nitrate N, 0.4 mM P, 3 mM K, 1.5 mM Ca, 0.8 mM Mg, 0.8 mM S, 0.3 mM Si, 25 μM Fe, 40 μM B, 3 μM Mn, 3 μM Zn, 4 μM Cu, 35 μM Cl, 0.1 μM Mo, and 0.1 μM Ni. Measured urea concentration was accurate, and no interferences were observed.

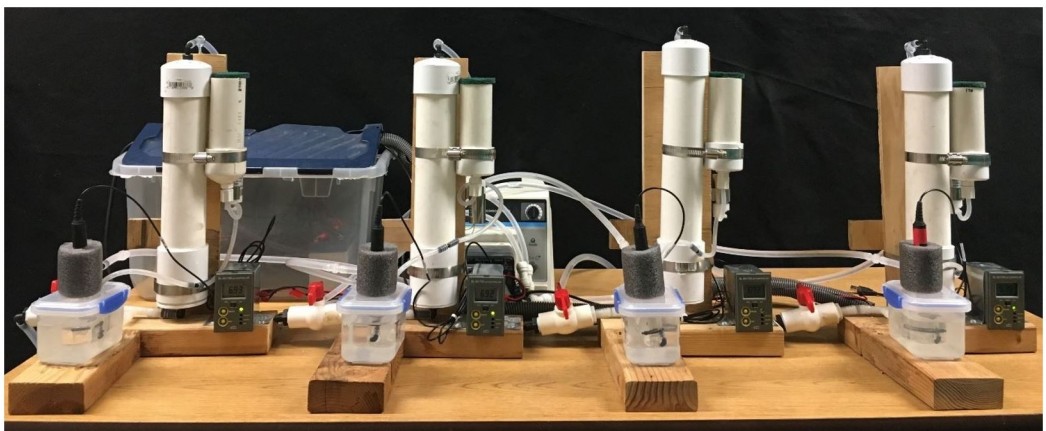

**Fig 3. System for measuring urea hydrolysis.** Recirculating columns filled with perlite were used to measure concentrations of urea and ammonium over 30 days.

**Fig 4. Changes in nitrogen form over time.** Concentrations of nitrogen in urea and ammonium over 30 days in recirculating columns (n = 2) controlled at pH 7 with sulfuric acid addition. Total N represents the sum of N from urea and ammonium. Error bars represent standard error, n = 2.

## Supporting information

**S1 File. Protocol for urea assay from protocols.io.**
(PDF)

**S2 File. Calibration curve and urea hydrolysis data.**
(DOCX)

## Author Contributions

**Conceptualization:** Noah James Langenfeld.

**Investigation:** Noah James Langenfeld, Lauren Elizabeth Payne.

**Writing – original draft:** Noah James Langenfeld, Lauren Elizabeth Payne.

**Writing – review & editing:** Bruce Bugbee.

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
