## [Decision Letter · Decision Letter 0]

5 Oct 2021

PONE-D-21-25118Colorimetric determination of urea using diacetyl monoxime with strong acidsPLOS ONE

Dear Dr. Langenfeld,

Thank you for submitting your manuscript to PLOS ONE. After careful consideration, we feel that it has merit but does not fully meet PLOS ONE’s publication criteria as it currently stands. Therefore, we invite you to submit a revised version of the manuscript that addresses the points raised during the review process. Please address all the points raised by the reviewers: I fully agree with them that the minor ,suggested changes will improve the overall quality of the manuscript

We look forward to receiving your revised manuscript.

Kind regards,

Giovanni Signore

Academic Editor

PLOS ONE

Journal Requirements:

Reviewers' comments:

Reviewer's Responses to Questions

**Comments to the Author**

1. Does the manuscript report a protocol which is of utility to the research community and adds value to the published literature?

Reviewer #1: Yes

Reviewer #2: Yes

2. Has the protocol been described in sufficient detail?

Descriptions of methods and reagents contained in the step-by-step protocol should be reported in sufficient detail for another researcher to reproduce all experiments and analyses. The protocol should describe the appropriate controls, sample sizes and replication needed to ensure that the data are robust and reproducible.

Reviewer #1: Partly

Reviewer #2: No

3. Does the protocol describe a validated method?

Reviewer #1: Yes

Reviewer #2: No

4. If the manuscript contains new data, have the authors made this data fully available?

Reviewer #1: Yes

Reviewer #2: Yes

**5. Is the article presented in an intelligible fashion and written in standard English?**

Reviewer #1: Yes

Reviewer #2: Yes

6. Review Comments to the Author

Reviewer #1: Dr. Langenfeld et al. has presented a work entitled as “Colorimetric determination of urea using diacetyl monoxime with strong acids”. The authors have addressed the raised queries of this reviewer in detail. In this relevance on the basis of this revised version of the “Lab Protocol” it may be considered for publication in the esteemed journal.

Reviewer #2: The protocol, that is not included in M&M section despite my suggestion in the first revision, but only available as supplementary material, is written in an unclear/uncorrect form:

• point nr 2:

Add 80 µl phosphoric acid. Unclear:

Which is the concentration of the Phosphoric Acid?

• point nr 3:

Prepare 18 Molarity (M) sulfuric acid by diluting 65.25 mL concentrated sulfuric acid up to 250 mL with deionized water.

This is uncorrect: 18 M Sulfuric Acid IS concentrated Sulfuric acid:

Concentrated Sulfuric Acid 98% (H2SO4) MW: 98.073

Density: 1.84 Approx. Strength: 96% Molarity(M): 18 Volume (mL) required to make 1000 mL of 1M solution: 55.

• The correct spelling of [mM] is Millimolar, not Milimolar

• Purity of reagents (thiosemicarbazide, diacetyl monoxime, phosphoric acid, ferric chloride) is not reported

7. PLOS authors have the option to publish the peer review history of their article (what does this mean?). If published, this will include your full peer review and any attached files.

Reviewer #1: No

Reviewer #2: No

---

## [Author Response · Author response to Decision Letter 0]

19 Oct 2021

Reviewer #2: 

The protocol, that is not included in M&M section despite my suggestion in the first revision, but only available as supplementary material, is written in an unclear/uncorrect form: This was done because we were following the template for a lab protocol laid out by PLOS One. Lab protocols are meant to have the protocol included only as supplementary information and a link placed in the materials and methods section with a DOI. No change has been made because of this reason.

• point nr 2: Add 80 µl phosphoric acid. Unclear: Which is the concentration of the Phosphoric Acid? The concentration has been added to the protocol.

• point nr 3: Prepare 18 Molarity (M) sulfuric acid by diluting 65.25 mL concentrated sulfuric acid up to 250 mL with deionized water. This is uncorrect: 18 M Sulfuric Acid IS concentrated Sulfuric acid: Concentrated Sulfuric Acid 98% (H2SO4) MW: 98.073 Density: 1.84 Approx. Strength: 96% Molarity(M): 18 Volume (mL) required to make 1000 mL of 1M solution: 55. 

You are correct, this was a mistake on our part and the correct concentration has now been included in the protocol.

• The correct spelling of [mM] is Millimolar, not Milimolar. This was a default setting on protocols.io. We have corrected the spelling.

• Purity of reagents (thiosemicarbazide, diacetyl monoxime, phosphoric acid, ferric chloride) is not reported. Purity has now been reported for all reagents.

---

## [Editor Report · Decision Letter 1]

26 Oct 2021

Colorimetric determination of urea using diacetyl monoxime with strong acids

PONE-D-21-25118R1

Dear Dr. Langenfeld,

We’re pleased to inform you that your manuscript has been judged scientifically suitable for publication and will be formally accepted for publication once it meets all outstanding technical requirements.

Kind regards,

Giovanni Signore

Academic Editor

PLOS ONE
---

## [Editor Report · Acceptance letter]

28 Oct 2021

PONE-D-21-25118R1 

Colorimetric determination of urea using diacetyl monoxime with strong acids 

Dear Dr. Langenfeld:

I'm pleased to inform you that your manuscript has been deemed suitable for publication in PLOS ONE. Congratulations! Your manuscript is now with our production department. 

Kind regards, 

on behalf of

Dr. Giovanni Signore 

Academic Editor

PLOS ONE